# Studies of the Effect of Seasonal Cycle on the Equatorial Quasi-Biennial Oscillation with a Chemistry-Climate Model

**Kiyotaka Shibata**

School of Environmental Science and Engineering, Kochi University of Technology, 185 Miyanokuchi, Tosayamada, Kami 782-8502, Kochi, Japan; shibata.kiyotaka@kochi-tech.ac.jp

**Abstract:** The effect of the seasonal cycle on the quasi-biennial oscillation (QBO) in the equatorial stratosphere was investigated using a chemistry-climate model (CCM) by fixing the seasonal cycle in CCM simulations. The CCM realistically reproduced the QBO in wind and ozone fields of a 30-month period in a climatological simulation (control run) under annually repeating sea surface temperature (SST) with a seasonal cycle. For the control run, four experimental simulations (perpetual runs) were made by fixing solar declination and SST on the 15th of January, April, July, and October, respectively, for about 20 years. In the three perpetual runs of January, July, and October, the QBO was maintained and persisted throughout the 20-year integration in spite of some small differences in period and amplitude among the three runs. On the other hand, the QBO in the perpetual April run began to weaken after about 15 years and the downward propagation of westerly wind stopped at about 20 hPa, resulting in the QBO's ceasing. The cause of this QBO disappearance is related to the evolution of the background mean flow in the lower stratosphere, which filtered out the parameterized gravity waves propagating upwards farther.

**Keywords:** quasi-biennial oscillation; seasonal cycle; chemistry-climate model; perpetual run

## 1. Introduction

In the tropics, the westerly and easterly winds alternate with about a 2-year interval, propagating downward from the upper stratosphere to the lower stratosphere (e.g., [1]). The period of this quasi-biennial oscillation (QBO) has a broad spectrum width from about 20 to 40 months, averaged at about 28 months (e.g., [1]). The basic mechanism of the QBO is explained by internal interactions between the background mean flow and the gravity (and equatorial) waves propagating upward from the troposphere (e.g., [2–4]). The QBO in the tropics is one of the dominant variabilities in the stratosphere and is known to affect the tropospheric and stratospheric atmosphere both in the tropics (e.g., [5,6]) and extratropics (e.g., [7,8]). The QBO has been analyzed to be modulated by, affected by, or correlated with several forcings or factors, such as the annual or seasonal cycle (e.g., [9–12]), El Niño or Southern Oscillation (e.g., [13,14]), huge volcanic eruptions (e.g., [15,16]), the 11-year solar cycle (e.g., [17,18]), and the Pacific Decadal Oscillation [19].

The seasonal modulation of the QBO is characterized primarily as the seasonal preference of the QBO onsets and phase transitions. For example, the west–east phase transitions of zonal wind at 44 hPa occur preferentially during northern hemisphere (NH) summers, while the east–west phase transitions at 20 hPa tend to occur around NH winters [18]. Likewise, the easterly and westerly transitions at 50 hPa tend to occur much more frequently during the NH late spring or summer than during the NH winter (e.g., [10,11]). A quantitative analysis of the JRA-25 Reanalysis for 30 years from 1979 to 2008 [20] showed that the westerly transition at 50 hPa occurred eight times for April–May–June (AMJ) and did not occur for June–August–September (JAS), and that the ratio of the easterly transition between AMJ and JAS was 7 to 1 [21]. This seasonal modulation is also referred to as annual or seasonal synchronization, seasonal locking, or phase alignment with the annual or seasonal cycle.

The cause of the annual synchronization of the QBO was ascribed to the annual cycle in the Brewer–Dobson circulation (BDC), i.e., the annual cycle in the upwelling in the tropical stratosphere (e.g., [10,12]). Numerical model studies demonstrated that the annual cycle in the BDC was responsible for the observed annual synchronization of the QBO (e.g., [22–24]). On the other hand, it was suggested that the seasonal variations of gravity waves associated with convective activity in the tropics play a key role in inducing the annual synchronization tendency of the QBO [21]. This is because the zonal wind tendency ($\partial U/\partial t$) of the QBO is positively and significantly correlated with the unresolved residual term and, at once, negatively and significantly correlated with the vertical advection term in the zonal momentum budget in the framework of the transformed Eulerian mean equation, while the remaining two terms, the meridional advection and resolved wave forcing, were scarcely correlated with $\partial U/\partial t$ [21]. This result indicates that the gravity wave forcing represented by the unresolved residual term largely contributes to drive the QBO [25] and that the vertical advection related to the BDC rather exerts the canceling force. In addition, the simulated ensemble QBOs by a chemistry-climate model (CCM) employing a temporally constant gravity wave source were analyzed to have very weak annual synchronizations [21], implicitly in line with the gravity wave forcing being a major cause of the annual synchronization of the QBO.

Because the cause of the annual synchronization of the QBO still remains unclear and debatable, in this study, a set of CCM simulations was utilized to better isolate the effect of the seasonal cycle by fixing climatological forcings at the middle month of each season. These months represent middle dates of the four seasons. More precisely, the simulations of a CCM were performed by switching in and out of the annual cycle of climatological forcings and comparisons were made focusing on the changes in the QBO between the simulation under time-evolving climatological seasonal forcings and the simulations under time-fixing forcings without the seasonal cycle. The latter simulations are referred to as perpetual season simulations (e.g., [26,27]). The use of a CCM for the simulation of the QBO is based on the fact that the ozone provides crucial effects on the QBO period [28]. The rest of this paper is organized as follows: Section 2 describes the model and simulation conditions. Section 3 presents the results of the simulated QBOs under the annually repeating climatological forcings and the fixed perpetual forcings of four seasons. Section 4 provides the discussion, and Section 5 provides conclusions.

## 2. Model Simulations

This study utilizes the CCM of the Meteorological Research Institute (MRI) of Japan (MRI-CCM), which is an update used in [29]. Specifications of the MRI-CCM are described in other papers [28,29] and references therein, so that only its dynamics module is briefly provided here. The dynamics module is an atmospheric spectral global model with triangular truncation, a maximum total wavenumber of 42 (T42, about 2.8° by 2.8° in longitude and latitude grid space), and 81 layers in the terrain-following eta-coordinate with a lid at 0.01 hPa (about 80 km). The layer thickness in the stratosphere is about 500 m. To spontaneously reproduce the QBO, parameterized non-orographic gravity-wave forcing by Hines [30] was incorporated, the source strength of which is temporally constant but latitudinally enhanced in a Gaussian form over the tropics. Furthermore, biharmonic ($\Delta^2$) horizontal diffusion was minimized between 100 and 10 hPa and vertical diffusion was set to zero in the middle atmosphere to retain sharp vertical wind shear in the QBO.

The MRI-CCM was integrated for about 20 years under climatological forcings, in which sea surface temperature (SST) and sea ice data is a monthly mean climatology of HadISST1 [31] from the 1870s to the 2000s. The solar irradiance was at solar minimum and the stratospheric aerosol was at background condition. Abundances of greenhouse gases and ozone-depleting substances are those in January 2000 of the B2 scenario of the second phase of Chemistry-Climate Model Validation Activity (REF-B2) [32]. The initial conditions were taken from the data of MRI-CCM [28] in January 2000 under the time-evolving forcings of the REF-B2 scenario. For this climatological simulation (control run),

four experimental simulations (perpetual runs) were made similarly for about 20 years under the conditions of perpetual four seasons in the NH winter, spring, summer, and autumn, in which SST or sea ice and solar declination were fixed on the 15th of January, April, July, and October, respectively, and solar radiation varied only diurnally.

It should be noted that the climatological SST or sea ice data annually repeat with the seasonal cycle and yet lack the effect of inter-annual variations. In addition, the perpetual runs do not include the effect of intra-seasonal variations in SST. The initial conditions were taken from the data of the control run on the first day of January, April, July, and October, respectively, in 2001. These runs are referred to as perpetual January, April, July, and October runs. Of the four perpetual runs, the perpetual January and July runs are also referred to as near-solstice runs and the perpetual April and October runs as near-equinox runs.

## 3. Results

### 3.1. Control Run

Figure 1 displays the time–height cross-section of zonal–mean-zonal wind anomalies averaged between 10° S and 10° N in the tropical stratosphere (100–5 hPa) for the control run for about 20 years. Anomalies mean the deviations from the annual cycle, i.e., the climatological monthly value is subtracted from an average for each month. The control run simulated a QBO for about a 30-month period and maximum amplitude at 20 hPa of about 18 ms$^{-1}$ with steeper westerly shear ($\partial U/\partial z > 0$) than easterly shear ($\partial U/\partial z < 0$), and these features approximately agree with observations [1,33]. Compared to the QBO simulated under time-evolving forcings of the REF-B2 scenario, the maximum amplitude at 20 hPa is nearly the same, but the period is slightly longer by about two months [28].

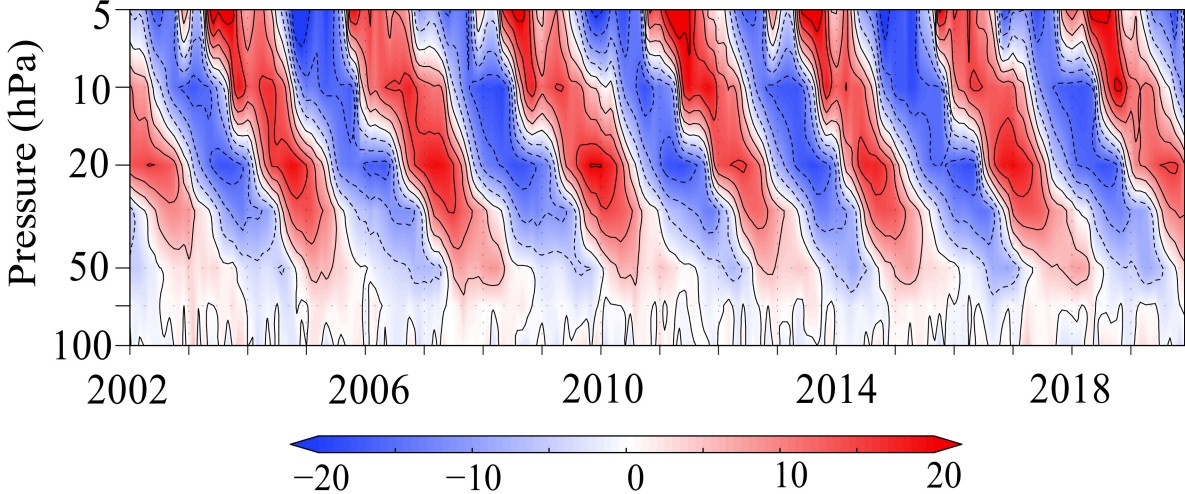

**Figure 1.** Time–height cross-section of zonal–mean-zonal wind anomalies (ms$^{-1}$) averaged between 10° S and 10° N in the tropical stratosphere (100–5 hPa) for the control run from 2002 to 2019. Contours interval is 5 ms$^{-1}$.

Figure 2 depicts the power spectrum of the zonal–mean-zonal wind averaged between 10° S and 10° N in the tropical stratosphere and mesosphere (100–0.1 hPa), which shows that the stratospheric QBO gradually diminishes with altitude in the upper stratosphere minimizing in the lower mesosphere. On the other hand, the annual cycle and the semiannual oscillation (SAO) become dominant in the upper stratosphere and mesosphere.

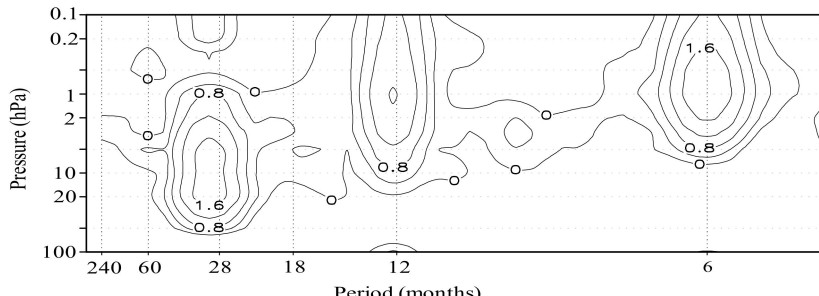

**Figure 2.** Power spectrum of zonal–mean-zonal wind from 100 to 0.1 hPa averaged between 10° S and 10° N. Values are displayed in a logarithmic scale of base 10 and units are in $m^2\,s^{-2}$. The contour interval is 0.4 $m^2\,s^{-2}$.

Figure 3 exhibits the time–height cross-section of the annual cycle of zonal–mean-zonal wind averaged between 10° S and 10° N in the tropical upper stratosphere and mesosphere (5–0.05 hPa) for the control run, in which the SAO is much more prominent than the annual component. The simulated SAO reproduced the observed features such that the first cycle beginning with the stratopause easterly phase in northern winter is larger than the second cycle beginning with the stratopause easterly phase in southern winter (e.g., [34]). The simulated SAO amplitudes of the first and second cycles are similar magnitudes to those in the reanalysis data in the lower mesosphere and stratosphere, respectively, while they are much smaller than those in satellite data in the middle and upper mesosphere [35,36].

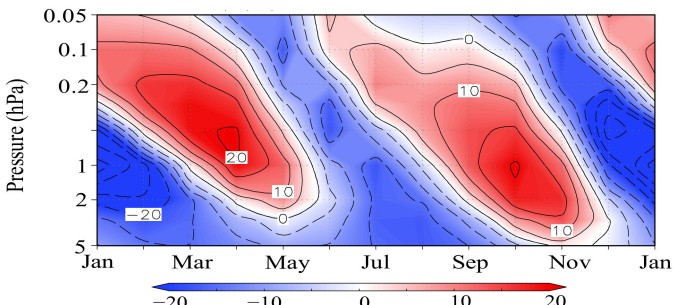

**Figure 3.** Time–height cross-section of the annual cycle of zonal–mean-zonal wind ($ms^{-1}$) averaged between 10° S and 10° N in the tropical upper stratosphere and mesosphere (5–0.05 hPa) for the control run. Contours interval is 5 $ms^{-1}$.

*3.2. Perpetual Runs*

Figure 4 depicts the time–latitude cross-section of zonal–mean-zonal wind at 10 hPa in the upper stratosphere for the perpetual January run, in which monthly averaged wind data are used. In the upper troposphere at 300 hPa, there blew persistently strong subtropical jets of 35–40 $ms^{-1}$ around 30° N in the NH and of about 30 $ms^{-1}$ around 45° S in the southern hemisphere (SH) with some intra-seasonal variations (not shown). On the other hand, at 10 hPa in the NH, the polar night jet around 70° N showed large intra-seasonal variations from 60 to 10 $ms^{-1}$, while there blew very weak easterly winds of less than 10 $ms^{-1}$ in middle and high latitudes in the SH. The large intra-seasonal variations, i.e., frequent sharp weakening of the northern polar night jet, indicate that there often occurred major sudden stratospheric warmings (SSWs) defined as the reversal of zonal–mean-zonal wind at 10 hPa in 60° N [37]. The daily averaged data more directly show the occurrences of SSWs (not shown).

Figure 5 displays the time–latitude cross-section of zonal–mean-zonal wind at 10 hPa in the upper stratosphere for the perpetual July run. In the upper troposphere at 300 hPa (not shown), the southern subtropical jet of about 35 $ms^{-1}$ is situated around 30° S, while the northern subtropical jet around 50° N is very weak (about 12 $ms^{-1}$). The polar night jet

at 10 hPa around 60° S is very strong and stable with intra-seasonal variations between 70 and 90 ms$^{-1}$.

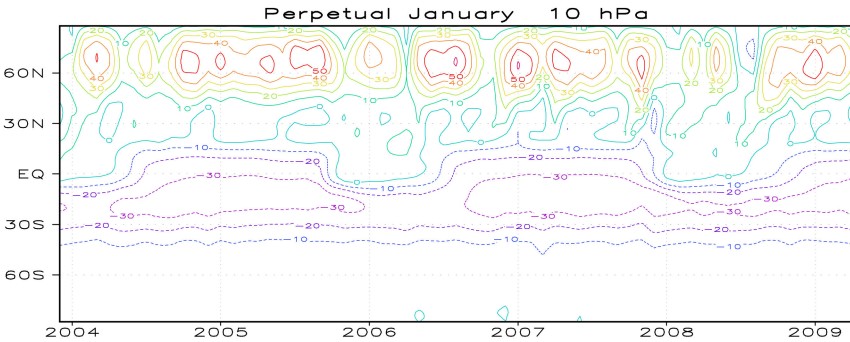

**Figure 4.** Time–latitude cross-section of zonal–mean-zonal wind (ms$^{-1}$) at 10 hPa in the upper stratosphere for the perpetual July run. Contours interval is 10 ms$^{-1}$.

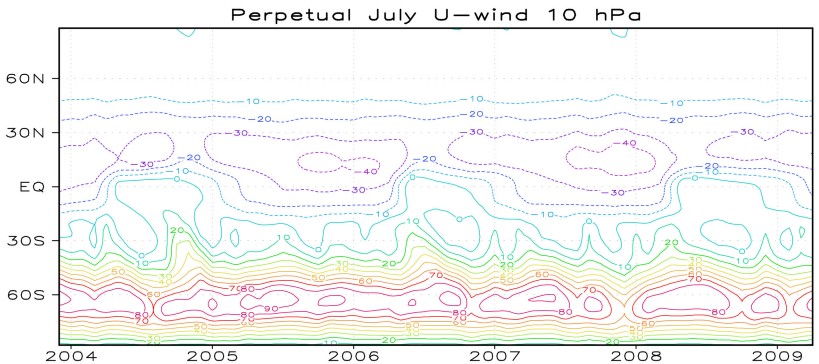

**Figure 5.** Time–latitude cross-section of zonal–mean-zonal wind (ms$^{-1}$) at 10 hPa in the upper stratosphere for the perpetual January run. Contours interval is 10 ms$^{-1}$.

Figure 6 shows the time–height cross-sections of zonal–mean-zonal wind anomalies (ms$^{-1}$) averaged between 10° S and 10° N in the tropical stratosphere (100–5 hPa) for the four perpetual runs. The three perpetual runs (January, July, and October) simulated the QBOs with similar periods and periods throughout the entire years, and their values at 20 hPa are 26 months and 15 ms$^{-1}$ for January, 25 months and 16 ms$^{-1}$ for July, and 26 months and 20 ms$^{-1}$ for October. On the other hand, the QBO in the perpetual April run began to diminish around 2013 and finally disappeared after 2016. This indicates that the QBO was simulated for the former 15 years, during which there were about 5 cycles. Following this, the downward propagation of the next weak westerly wind (6th in Figure 6c) virtually stopped at 20 hPa and after that there persisted weak easterly anomalies below 30 hPa and weak westerly anomalies between 10 and 20 hPa. During the former 15 years, the vertical profile of the QBO amplitude also differs from the other QBOs, which maximized between 10 and 20 hPa for both the westerly and the easterly winds. That is, below 10 hPa in the perpetual April run, the maximum values appeared at 30 hPa for the westerly winds, being in contrast to the maximum values at 20 hPa for the easterly winds. To be specific, the period and amplitude of the QBO at 20 hPa in the perpetual April run is 28 months and 8 ms$^{-1}$.

Figure 7 exhibits the power spectrum of zonal–mean-zonal wind from 100 to 0.1 hPa, averaged between 10° S and 10° N for the four perpetual runs. The power spectrum in the period ranges from about 20 to 40 months corresponding to the QBO power. It is natural that the four perpetual runs did not simulate the SAO in the mesosphere, because one of the driving forces of the SAO stems from the semiannual meridional flow across the equator between the two hemispheres (e.g., [38,39]), which did not exist in the perpetual runs. The QBO power spectrum is similar among the three perpetual runs (January, July, and October)

below the stratopause though there are slight differences for the center periods as stated above. Compared to these three perpetual runs, the QBO power in the perpetual April run is much smaller in the stratosphere.

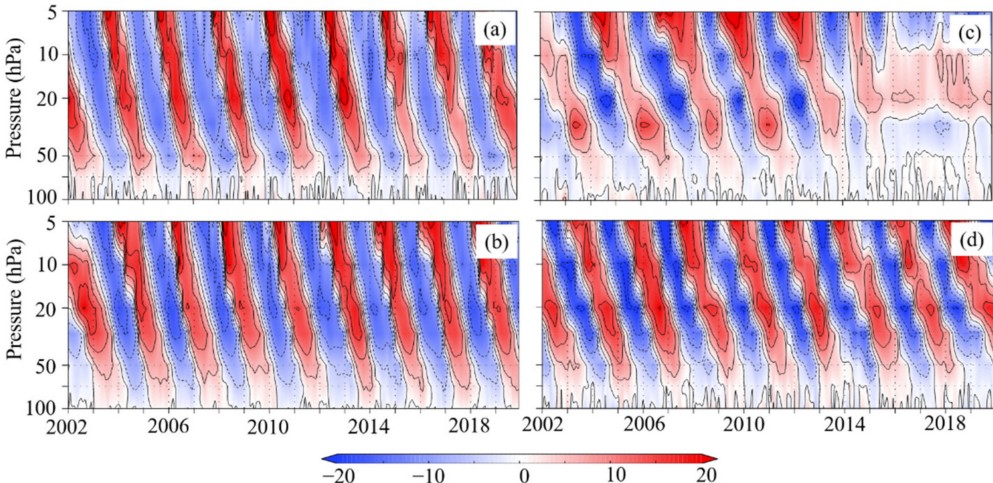

**Figure 6.** Power spectrum of zonal–mean-zonal wind from 100 hPa to 0.1 hPa averaged between 10° S and 10° N for the perpetual runs of (**a**) January, (**b**) July, (**c**) April, and (**d**) October. Values are displayed in a logarithmic scale of base 10 and units are in m² s⁻². The contour interval is 0.4 m² s⁻².

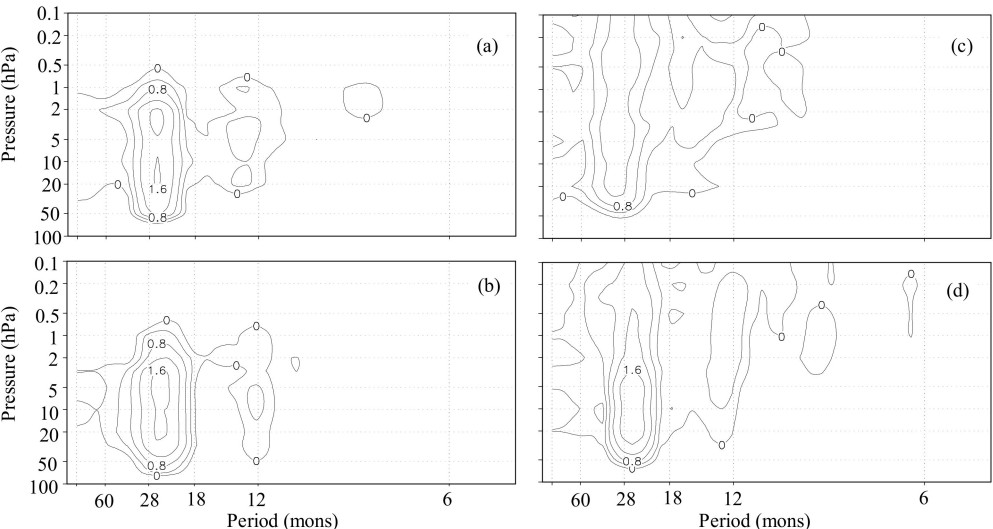

**Figure 7.** Time–height cross-sections of zonal–mean-zonal wind anomalies (ms⁻¹) averaged between 10° S and 10° N in the tropical stratosphere (100–5 hPa) for the perpetual runs of (**a**) January, (**b**) July, (**c**) April, and (**d**) October. Contours interval is 5 ms⁻¹.

On the other hand, above the stratopause, there are conspicuous differences in the QBO power between the near-solstice runs (January and July) and the near-equinox runs (April and October). In the near-solstice runs, the QBO power decreases very steeply with altitude above 2 hPa, resulting in virtually no QBO power around 0.5 hPa as in the control run. In contrast, the QBO signal protruded beyond the stratopause in the mesosphere without weakening in the near-equinox runs. Figure 8 displays the time–height cross-sections of zonal–mean-zonal wind anomalies (ms⁻¹) averaged between 10° S and 10° N in the tropical upper stratosphere and mesosphere (10–0.1 hPa) for the control run, the perpetual runs of January, and October, and Figure 9 shows those for the perpetual runs of April and October. It is evidently exhibited that the QBOs in the control run and the near-solstice runs sharply declined above 2 hPa and scarcely protruded into the mesosphere (Figure 8). On the other hand, the QBOs in the near-equinox runs protrude beyond the stratopause into

the mesosphere (Figure 9), though the QBOs minimize just below the stratopause except for the easterly wind in the perpetual April run. The difference of the QBOs in the mesosphere among the control, near-solstice, and near-equinox runs seems to come partly from the difference of the background mean zonal winds in the upper stratosphere. The background mean zonal winds below the stratopause at 2 hPa were easterlies of about 27 and 16 ms$^{-1}$ in the perpetual runs of January and July, while they were westerlies of about 11 and 10 ms$^{-1}$ in the perpetual runs of April and October (not shown). These features were approximately similar to the phases of the SAO in corresponding months of the control run (Figure 3).

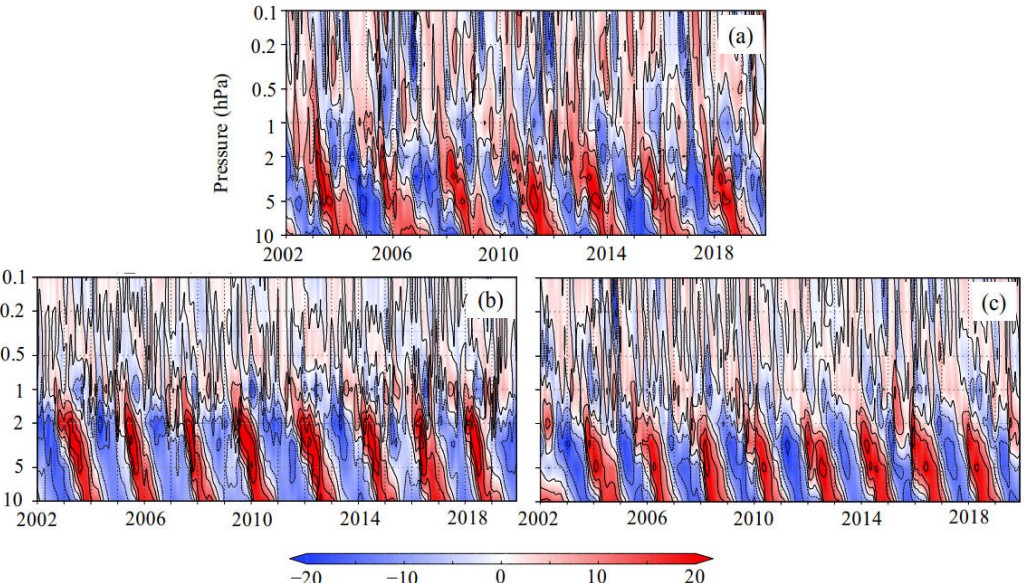

**Figure 8.** Time–height cross-sections of zonal–mean-zonal wind anomalies (ms$^{-1}$) averaged between 10° S and 10° N in the tropical upper stratosphere and mesosphere (10–0.1 hPa) for (**a**) the control run, the perpetual runs of (**b**) January, and (**c**) July. Contours interval is 7 ms$^{-1}$.

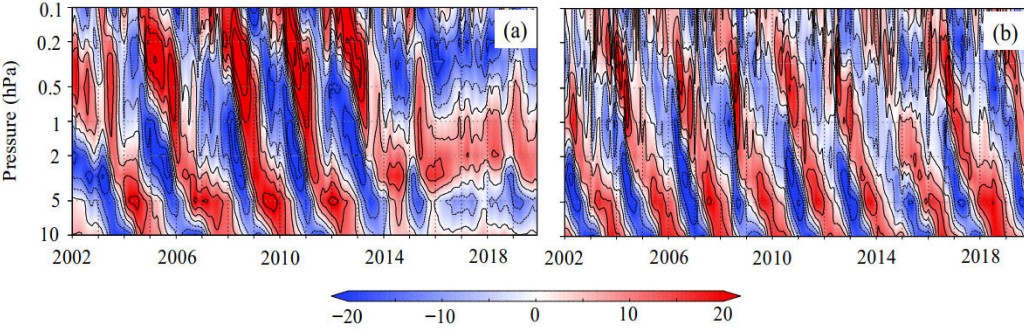

**Figure 9.** Time–height cross-sections of zonal–mean-zonal wind anomalies (ms$^{-1}$) averaged between 10° S and 10° N in the tropical upper stratosphere and mesosphere (10–0.1 hPa) for (**a**) the perpetual runs of (**a**) April and (**b**) October. Contours interval is 7 ms$^{-1}$.

## 4. Discussion

In this section, the forcing of the QBO is investigated with a focus on the circumstances, in which the QBO disappeared after about 5 cycles in the perpetual April run. The driving forces of the QBO in the MRI-CCM are largely due to parameterized gravity waves launched at the lowest level [40] and resolved waves propagating from the troposphere. The parameterized gravity-wave forcing is referred to as gravity-wave forcing (GWF), and the resolved wave forcing is represented by the Eliassen–Palm flux divergence (EPD). Of these two forcings, GWF plays a more crucial role in reproducing the QBO in the MRI-CCM, because there appeared no QBO without GWF. Figure 10 shows the time series of the QBO components of zonal–mean-zonal wind, EPD, GWF, and combined forcing (EPD + GWF)

at 20 hPa for the control run, the perpetual runs of January, and April from 2010 to 2019, in which a band-pass Lanczos filter [41] was applied to derive the QBO components defined as those between a 20- and 40-month period.

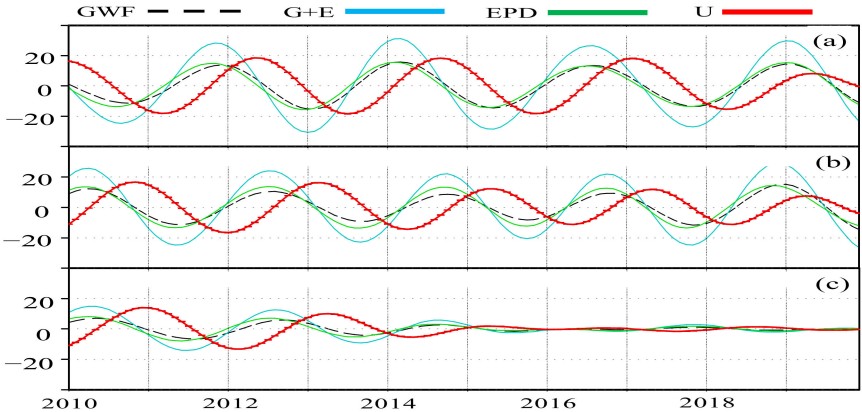

**Figure 10.** QBO components of zonal–mean-zonal wind (red) and parameterized gravity wave forcing (GWF, black dashed line), resolved wave forcing (EPD, light green), and combined forcing (G + E, cyan)) at 20 hPa for the (**a**) control run, perpetual runs of (**b**) January, and (**c**) April from 2010 to 2019. All of the quantities are averaged between 10° S and 10° N. Units are in ms$^{-1}$ for wind, and $10^{-2}$ ms$^{-1}$ day$^{-1}$ for forcings.

EPD and GWF at 20 hPa are nearly of the same amplitude and in-phase in each run, even in the decaying phase (2013–2016) of the perpetual April run. The combined forcing of EPD + GWF is advanced by a quarter cycle to zonal wind and its magnitude is approximately proportional to the zonal-wind amplitude in these three runs, demonstrating that it is the driving force of the QBO. These features among zonal wind, EPD, and GWF at 20 hPa are similarly seen in the perpetual runs of July and October (not shown). In other altitudes, on the other hand, the phase and magnitude of GWF differ from those of EPD, with the altitude difference from 20 hPa, particularly, in the lower stratosphere around 70 hPa, where the zonal wind QBO is very weak.

While the QBO components were recurring without suspension in the forcings and zonal wind, with some variations both in the control run and the perpetual January run, both EPD and GWF began to decline around 2012, prior to the weakening of zonal wind after 2013, in the perpetual April run. Decline of the vertical velocity QBO also may well be seen following the weakening of zonal wind QBO. Figure 11 shows the residual mean vertical velocity anomalies and zonal–mean-zonal wind at 20 hPa for the control run, the perpetual runs of January, and April from 2010 to 2019. The vertical velocity QBO is retarded by about a quarter cycle to zonal wind QBO [42,43] for the three runs, implicitly indicating that the decline of the vertical velocity QBO after about 2013 in the perpetual April run is not a cause but a result of the weakening of zonal wind.

Although the cause of the weakening of zonal wind QBO after 2013 in the perpetual April run has not yet been identified, the decline of GWF in the lower stratosphere is certainly related to this weakening. This is because EPD weakened much more gradually than GWF in the perpetual April run (not shown). Indeed, GWF was smaller than EPD in the stratosphere and troposphere, but the decline of GWF made the total combined forcing of EPD + GWF smaller than a certain critical value required for the generation of the QBO. The effect of the Coriolis torque on $\partial U/\partial t$ was smaller than EPD and GWF as in [21]. Further, v* declined more gradually than GWF, similarly to w* (Figure 11). Accordingly, the decline of GWF acted as a trigger for the diminishing and disappearance of the QBO.

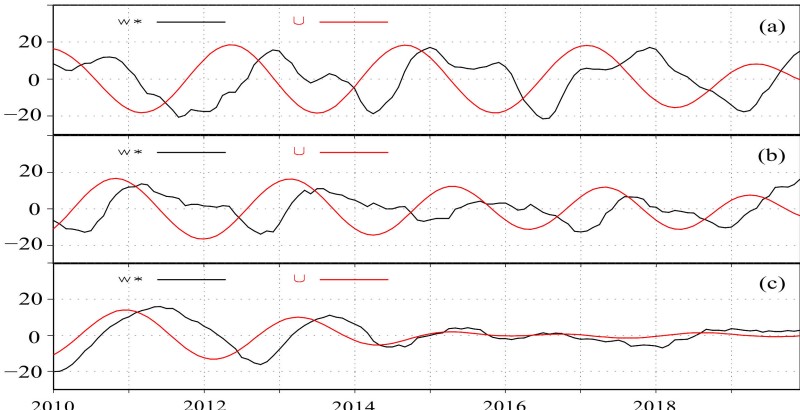

**Figure 11.** Zonal–mean-zonal winds (QBO components) (red) and residual mean vertical velocity anomalies (w *) (black) at 20 hPa for the (**a**) control run, perpetual runs of (**b**) January, and (**c**) April from 2010 to 2019. All of the quantities are averaged between $10°$ S and $10°$ N. Units are in $\text{ms}^{-1}$ for zonal wind and $10^{-5}\ \text{ms}^{-1}$ for w *.

Figure 12 exhibits the QBO components of GWFs from the upper troposphere to the lower stratosphere in the tropics for the perpetual runs of January and April from 2010 to 2019. In the perpetual January run, the amplitude of GWF became steeply larger, with altitude in a pressure range of 300–70 hPa, and its variation among the QBO cycles was very small in the lower stratosphere (70 hPa). This indicates that there is nearly constant and persistent acceleration (and deceleration) of zonal wind due to GWF in the stratosphere. In the perpetual April run, on the other hand, GWF apparently declined during 2013–2016 in the lower stratosphere and virtually disappeared after that. This GWF behavior at 70 hPa is in contrast to those in the upper troposphere at 100 and 300 hPa, where the amplitudes of GWF are of similar magnitude to the perpetual January run. The sudden GWF decline occurring at 70 hPa around 2013 indicates the parameterized gravity waves could not propagate upward beyond 70 hPa. One possible cause of this decline is some changes in the background mean flow in the tropical lowermost stratosphere. However, there did not seem to be prominent changes there around 2013. To scrutinize the mechanism of the decline and disappearance of the QBO in the perpetual April run is beyond the scope of this study and thus constitutes a future work. As stated before, the QBO effect extends to the extratropics in the stratosphere or troposphere (e.g., [1,44,45]) through the modulation of the planetary wave propagation (e.g., [46–48]). However, investigating the QBO effects in the extratropics in the perpetual runs is also a future work, because this study is only aimed at focusing on the impact on the QBO itself.

In the control run, a major driving force was imposed as a climatological SST, which did not include inter-annual variations. Furthermore, in the perpetual run, the effect of intra-seasonal variations in SST was excluded. Hence, SST forcing and the CCM response may be more or less underestimated both in the control and perpetual runs. To set more realistic forcing, it is preferable to impose observed SST with inter-annual variations (e.g., [49]) or to incorporate interactive SST (e.g., [44,50]) through coupling with an ocean model. To utilize such an elaborated setting of SST is also a future work.

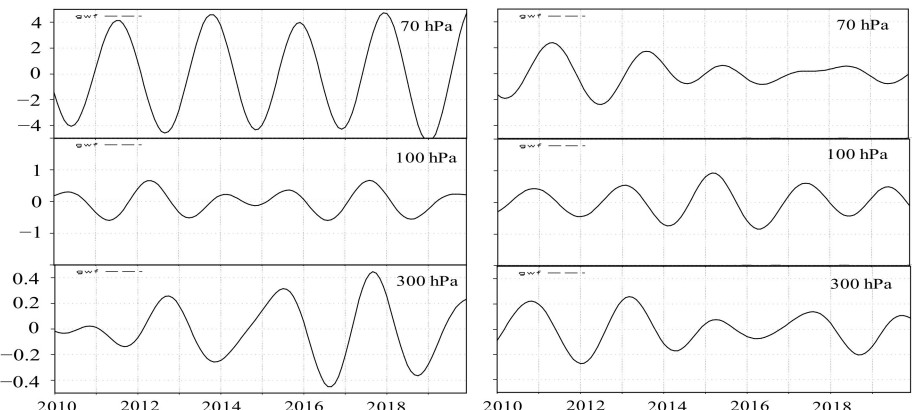

**Figure 12.** QBO components of GWFs at 70, 100, and 300 hPa for the (**left**) perpetual runs of January, and (**right**) from 2010 to 2019. All of the quantities are averaged between $10°$ S and $10°$ N. Units are in $10^{-2}$ ms$^{-1}$ day$^{-1}$.

## 5. Conclusions

Under annually repeating climatological forcings, i.e., solar and SST forcings, MRI-CCM simulated the QBO for about a 30-month period and with about an 18 ms$^{-1}$ maximum amplitude at 20 hPa, which are approximately similar to observations. Even under the four perpetual runs of January, April, July, and October, in which external forcings were fixed to the climatological forcings at each respective month, MRI-CCM also reproduced the QBOs possessing similar features to the QBO in the control run except for the latter several years in the perpetual April run. However, these simulations do not necessarily indicate that the effect of the seasonal cycle on the QBO is small. This is because the source of GWF in the CCM is not time-evolving but constant, resulting in GWF depending mostly on the background mean flow. Hence, to more rigorously investigate the effect of the seasonal cycle, the source of GWF should somehow include the effect of convective activity responsible for gravity wave generation in the troposphere.

Prominent effects of the seasonal cycle appeared in the tropical mesosphere, where the SAO is the dominant variation. In the near-solstice runs, the QBOs were confined below the stratopause and hardly protruded into the mesosphere, which was qualitatively in parallel with the control run. On the other hand, the QBOs in the near-equinox runs protruded beyond the stratopause into the mesosphere. The difference of the QBOs in the mesosphere among the control, near-solstice, and near-equinox runs seems to be related to the difference of the background mean zonal winds in the upper stratosphere. The background zonal winds below the stratopause at 2 hPa were easterlies of about 27 and 16 ms$^{-1}$ in the near-solstice runs of January and July, as well as westerlies of about 11 and 10 ms$^{-1}$ in the near-equinox runs of April and October, the phase and magnitude of which were approximately similar to those of the SAO in the corresponding months of the control run.

**Funding:** This research received no external funding.

**Institutional Review Board Statement:** Not applicable.

**Informed Consent Statement:** Not applicable.

**Data Availability Statement:** The data presented in this study are available on request from the corresponding author.

**Acknowledgments:** The standard and experimental simulations of MRI-CCM were performed with the supercomputer system (NEC SX-ACE) of the National Institute for Environmental Studies, Japan.

**Conflicts of Interest:** The author declares no conflict of interest.

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
