# Peer review of "Studies of the Effect of Seasonal Cycle on the Equatorial Quasi-Biennial Oscillation with a Chemistry-Climate Model"

_climate, doi:10.3390/cli10070099_

Round 1
Reviewer 1 Report
Authors have addressed all my points that I raised whilst reviewing the original version. The paper is now ready for publication.
Reviewer 2 Report
I have reviewed the documents related to the paper.
The authors have answered all my comments.
Author Response
Please see the attchment.

Reviewer 3 Report
The author has been responsive both to my general comments and also to the specific sticky-note comments included in my original review.
Author Response
Please see the attchment.

Reviewer 4 Report
Please look at the attachment file.

Author Response
Please see the attchment.

Round 2
Reviewer 4 Report
Please look at the attached file.

Author Response
Please see the attachment.

This manuscript is a resubmission of an earlier submission. The following is a list of the peer review reports and author responses from that submission.
Round 1
Reviewer 1 Report
This is a very thorough and transparent study on QBO modelling. As the authors rightly mention in the Introduction, the QBO is correlated or linked in one form or the other with various climatic parameters (Lines 28-32). In order to make the paper more attractive for non-specialists, it might be worth adding additional and more detailed examples, how the QBO is though to be linked to climate in the troposphere. A good paper for this is Petick et al. 2011: doi:10.1029/2011JD017390. Another one that looked at the polar vortex is Camp & Tung 2007, https://journals.ametsoc.org/view/journals/atsc/64/4/jas3883.1.xml
The author is also using the term „seasonal change“. It may not be totally clear to everyone what this means in this case. Could it be simplified to „seasonal influence“ or something more intuitive? Some readers might think it is about longterm changes of a particular season over 100 or 1000 years.
The english language and figures are of high quality. Figure 10 looks a bit distorted.
Reviewer 2 Report
It is an interesting paper and shows a relevant topic. However, the methodology and especially the model simulations are very scarce. The author only describes the model without giving mathematical details.
The author refers that the specifications of MRI-CCM can be found in the literature. So this assumes that this submitted manuscript is only an application of the MRI-CCM model. If this is correct, then there is no scientific contribution in the submitted manuscript.
For example, in lines 81-82, it is said that "latitudinally enhanced in a Gaussian form over the 82 tropics". Where is this Gaussian condition verified? How can it be shown that it is not a Geometric or Exponential relationship?
Another example of the imprecise presentation of the model. In lines 83-85 it is said that the horizontal biharmonic diffusion is minimized down to 10hPa. The obligatory question, how is this obtained in the model? Without a mathematical equation it is very difficult to know how this process is accomplished. Something similar could also be said about the "transpor of chemical species".
The limitation in the description of the model is critical. The description of the Control run condition is null. In relation to the fact that the Control run is of great importance, it should be detailed in the text (mathematically).
Figure 2 shows directly the results of Power spectrum, however previously in the text nothing is said about it. Lines 120-124 only describe figure 2, but there are no further details on how it was obtained. The same happens with figure 7, it is not possible to know where it is obtained from.
The absence of this detailed methodological description makes the results questionable. For example Lines 112-113. What is the relationship between the Control run and "climatological monthly value is subtracted from an average for each month" At what point in the model should this subtraction be made? None of this is specified in the methodology section.
Lines 192-193. It is evidently exhibited that the QBOs in the control run and 192 the near-solstice runs sharply declined above 2 hPa and scarcely protruded into the 193 mesosphere. It does not seem obvious, as it is not known whether it refers to figure 8 or figure 9.
No vertical scale in figures 10, 11, 12 are assumed to be hPa.
In general, the manuscript is an application of a model that is undescribed and only the results obtained are described. This paper does not consider any innovative scientific contribution.
Reviewer 3 Report
This paper is about a modeling study of the Quasi-Biennial Oscillation (QBO) that takes place in the equatorial stratosphere and particularly the seasonal affect on the QBO, as investigated using a “chemistry-climate model.” Since the seasonal affect on the QBO is already well known (literature cited by the author), the reader imagines that this study is aimed at identifying phenomena that in principle could account for the seasonal affect (I say “in principle” because this is entirely a modeling study).
The paper seems to be written for an audience that already is conversant in the QBO and its significance, and appears written for an audience that already understands the CCM. It needs to be “translated” to a broader audience, one that may not know what this oscillation is or why it is important.
Along the same lines, the author needs to state the purpose, rationale and significance of this study so that a reader will better understand why he did the study in the first place and what it means. The reader also needs to know exactly what is new and significant in this study, which seems not to be stated.
A list of acronyms is needed to assist a reader's understanding.
For the average reader of Climate, I believe this paper in its present form will be hard to understand without these refinements. I have entered my specific queries in comments and highlighting within the original PDF version that I reviewed. I will not repeat those comments here, but instead ask that this edited copy be transmitted to the author to assist in any revisions he may choose to make.
Among the most significant questions that the work raises in my mind are:
1) Why use, and emphasize (in the title) a Chemistry Climate Model (CCM). I see no result here that employs any chemical parameters of atmospheric gases or anything else.
2) What is the rationale for distinguishing between “experimental” and “control” groups? This usage differs from the usual usage of these terms in empirical studies. How do the results from control groups validate or clarify results from the experimental group?
3) What is the significance of the results reported here? How do they help us to understand the climate (weather) phenomena evaluated?
4) Related, how do these results compare with empirical observations of the same phenomena?
5) How, therefore, do these modeling studies advance our understanding of the OBO and its causes? What is new here?
I have the impression that the model used is well-developed, haveing been used for many years, but am not sure why the chemistry aspect is emphasized or needed, since none of the results seem to rely upon chemical properties of atmospheric gases or interactions.
The scientific work is sound and this paper could be published with suitable revisions.
